# Developing Healthy Lifestyle Behaviors in Early Age—An Intervention Study in Kindergartens

**DOI:** 10.3390/nu15112615

**Published:** 2023-06-02

**Authors:** Ronit Jakobovich, Elliot M. Berry, Asia Levita, Diane Levin-Zamir

**Affiliations:** 1Department of Health Promotion, School of Public Health, Tel Aviv University, Tel Aviv 6209804, Israel; 2Department of Human Nutrition and Metabolism, Braun School of Public Health, Hebrew University Hadassah Medical School, Jerusalem 9103102, Israel; 3Faculty of Science and Technology Education, Technion, Haifa 3498838, Israel; asialevita1@gmail.com; 4School of Public Health, University of Haifa, Haifa 3498838, Israel; diamos@inter.net.il; 5Department of Health Education and Promotion, Clalit Health Services, School of Public Health, University of Haifa, Tel Aviv 6209804, Israel

**Keywords:** early childhood, parents, healthy nutrition, kindergarten intervention, Mediterranean lifestyle change, mixed-methods, mathematical-logical thinking, health literacy

## Abstract

Childhood obesity prevention is a leading public health challenge requiring the adoption of healthy lifestyles at an early age. We examined how the kindergarten environment can promote eating sensibly, drinking water and becoming physically active. The effects of an intervention program among 42 Israeli kindergartens (1048 children, aged 4–6) whose teachers participated in a health education training program were compared to 32 kindergartens (842 children) whose teachers did not undergo this training program. An eight-month intervention program focused on knowledge/mathematical/logical/critical thinking, self-regulation/control acquisition, and sensible decision-making abilities. We hypothesized that nutrition and physical-exercise-oriented intervention programs, combining knowledge/mathematical logical thinking, would positively impact the quality of children’s mid-morning snack and water consumption, their ability to express feelings following physical exercise, and the adoption of healthy lifestyles at home. The quality of mid-morning snacks and water consumption were observed in both groups pre- and post-intervention. Qualitative interviews documented children’s subjective feelings following physical exercise. A significant improvement (*p* < 0.001) was observed in the mid-morning snacks composition and in water drinking habits in the intervention group; 80% of children offered a physiological explanation regarding energy expenditure processes following intense physical exercise. In conclusion, kindergarten interventions implemented by trained teachers can promote adoption of health behaviors necessary for obesity prevention.

## 1. Introduction

### Background and Purpose

The prevention of obesity is one of the foremost challenges of public health. In developed countries, more than a quarter of children of kindergarten age are overweight or obese, and the prevalence is increasing [1], leading to long-term effects on morbidity in adolescence and adulthood [2,3]. Already by the age of three, poor eating habits have been observed, including consuming foods and drinks high in energy, with few fruits and vegetables [4,5]. Excessive screen-time has been shown to encourage eating snacks and sweet drinks, along with reduced physical activity [6]. The long-term consequences of childhood obesity lead to adult obesity and chronic diseases (such as the metabolic syndrome, some cancers, joint complications and psycho-social problems) [7]. This emphasizes the importance of acquiring healthy lifestyle habits at an early age, when behavior patterns are shaped and thus are more likely to be maintained throughout the life course [8,9]. At this early age, the ability to self-control and direct behaviors (self-regulation) develops, affecting all areas of behavior [10]. Therefore, interventions to influence healthier lifestyles may be more effective if implemented in early childhood, before unhealthy choices become fixed in an individual’s lifestyle [11].

The recommendations for childhood nutrition, physical activity and reduction of screen-time are as relevant as ever, as expressed by expert teams in the Position Statement on Childhood Obesity from the Global Federation of International Societies of Padiatric Gastroenterology, Hepatology and Nutrition (FISPGHAN) [12].

Based on scientific evidence, the Mediterranean diet and lifestyle provide major health benefits for preventing chronic non-communicable diseases [13]. Fruits and vegetable consumption are an integral part of this diet and may be encouraged from an early age [4].

Recent studies have emphasized the importance of meeting specific health literacy (HL) needs of children to gain the skills, knowledge and motivation to access and understand age-appropriate health-related information in order to make informed decisions in everyday life and to develop healthy lifestyle behaviors [14].These skills engage and empower young children and bring their lived experiences to encourage learning that is meaningful in their world. Children learn effectively by experimentation through structured-play-based activities which engage them in their design, implementation and evaluation—all by active, experiential participation in situations that necessitate applying knowledge that results in behavioral changes [15].

Social Learning Theory (SLT) [10] stresses great importance on the knowledge and skills that are acquired in the kindergarten setting, where most children spend the majority of their day [16,17]. Thus, children act within a social context and determine and control behaviors by themselves (self-regulation) through cognitive processes, as well through observing and imitating others in their environment [10]. Therefore, early childhood learning should include experiences and thinking activities while interacting with others and improving language proficiency [18]. Rich language and logical mathematical skills (e.g., quantity and size, a lot, a little, more/less, counting, ordering, graphic representation, comparing groups, the concepts of time, estimation and measurement), promote the skills of thinking and learning processes. These enable children to acquire knowledge, implement, plan, develop insights, foster precise thinking, analyze their actions, solve problems, develop goals and act with self-control and self-management [19,20,21]. Constructivist teaching (construction of knowledge) allows teachers to provide opportunities for children to use their cognitive skills to raise a personal issue or problem in an authentic context while being supported by classroom and peer-group discussions and appropriate activities [22].

Childhood obesity prevention has catalyzed many important studies, many of which focus on measurement tools for body weight and their validation and fidelity as outcome measures [23]. Yet, the importance of observing health behaviors and, moreover, analyzing children’s verbal responses are critical for capturing their experience and the significance that they express regarding intervention participation.

Programs have been tested that are based on the role of the kindergarten teacher in conducting the intervention, focusing on physical activity and motor ability outcomes while also identifying gender and socio-economic associations with positive responses to the activity [24,25].

Studying children’s attitudes towards healthy eating is of utmost importance, even when the intervention includes directly providing nutritious foods [26].

The earlier the intervention program begins, the greater the improvement attained over time [18,27]. A review of intervention programs among children aged 3–6 years recommended multiple pursuits, including growing vegetables, joint activities for parents and children, and changes in the learning environment [4,28]. An increasing number of researchers have extended lifestyle modification interventions to groups for overweight and obese preschool-age children, producing small but significant changes in weight which may have long-term effects [3]. These interventions generally focus on modifying behaviors related to energy balance, including nutrition (e.g., increasing fruits/vegetables, decreasing high fat foods and sugar-sweetened beverages), physical activity and sedentary habits [29].

The aim of this study was to examine how to utilize the kindergarten environment to create behavioral changes that help prevent obesity by eating a sensible diet, drinking water and engaging in physical activity. We hypothesized that nutrition and physical-exercise-oriented intervention programs, including meaningful activities combining knowledge and mathematical logical thinking, would impact positively on:The quality of the children’s mid-morning snack and water consumption.Children’s ability to explain how they feel about their bodies following physical exercise.Indirectly, parent involvement through the children, indicated by changing the quality of the children’s mid-morning snack and water consumption—thus showing that the children were acting as “agents of change”.

## 2. Materials and Methods

### 2.1. Setting

This study, conducted in 2012 and partially described was carried out among children aged four to six years divided into intervention and comparison kindergartens. The study targeted two populations: (a) *kindergarten teachers*: to train them about healthy lifestyles, to design an educational intervention for the children based on proper nutrition (energy intake) and physical activity (energy expenditure), and to encourage the integrative application of mathematical-logical language to themes and activities associated with daily life [30,31]; (b) *kindergarten children*: to introduce these lifestyle principles through the educational environment, and to promote their body awareness by talking about their feelings following physical exercise [32,33].

The intervention model in this study was based on SLT (Figure 1). The research focused on changes in the children’s habits facilitated by the teachers, the children themselves, the kindergarten environment and the parents as well. The intervention program expanded the attitudes and abilities of the teachers to devote more time to HL, to teach healthy behaviors and to use logical-mathematical language to acquire skills regarding nutrition and physical exercise, important for staying healthy as well as influencing the home environment.

#### 2.1.1. For Kindergarten Teachers

The intervention included a 56-h course on continuing education conducted throughout the entire school year. The sessions dealt with four areas (Table 1): (1) Health education and promotion; (2) Nutrition; (3) Physical activity; and (4) Imparting skills for mathematical and logical thinking.

The syllabus was based on SLT theory concerning language development and concepts necessary for acquiring and applying knowledge. Following a general overview of early child growth and development, specific components of the program included:Adopting sensible eating and drinking habits through menu diversification towards the Mediterranean diet pattern, prudent food choices and promoting self-control [34,35]. Emphasis was placed on increasing fruit, vegetable and water consumption and improving the quality of the mid-morning snack brought to school [36,37,38].Encouraging physical exercise in line with the Physical Exercise Curriculum for Kindergartens, encouraging teachers to be active and develop basic motor skills in the children such as walking, running, twisting in different directions, at different speeds, jumping and landing, rolling and balancing the body, shooting at targets, ball control and dribbling [39].The use of logical-mathematical language skills related to measurements such as: counting, concepts, quantity, time, speed, strength, representations in diagrams and monitoring charts, classification processes, comparisons, cause-effect relationships, drawing relevant conclusions, reasoning, and problem solving [40,41]. The educational environment illustrated these topics through wall diagrams, charts and computer graphics. Teaching modules encouraged activity in an experiential manner such as: using food groups to build food pyramids, preparing salads, documenting the amount of vegetables, fruits, and water consumed, as well as encouraging more physical exercise by counting and measuring steps.

#### 2.1.2. For Kindergarten Children

The program included significant activities integrating the knowledge and use of mathematical concepts such as: counting, more or less, drawing comparisons and classifying—all skills that enable children to develop critical abilities, tools for self-regulation/control, the ability to make sensible decisions, drawing conclusions and acquiring behaviors related to self-understanding “what is important for my body”; “what and how much do I eat and drink”—as a way of teaching a healthy lifestyle. To increase water consumption, they recorded “how much water did I drink today?” by placing their names in the class wall diagram according to the quantity of water drunk. The children calibrated their bottles, full, half and empty. Thus, they collected data, presented them and drew conclusions: “how many children drank a whole bottle of water today and how many drank half?”

### 2.2. Participants and Assignment of Groups

The participating school regions were based on convenience sampling based on Ministry of Education recommendations. The assignment to the intervention and comparison groups was not conducted randomly as the intervention plan was based on voluntary participation in in-service training for the teachers. The intervention group consisted of children from 42 kindergartens—1048 children (584 boys, 464 girls)—whose teachers volunteered to participate in the year-long experimental ‘health education’ intervention program. The comparison group was comprised of 842 children (475 boys, 367 girls) from 32 kindergartens whose teachers chose other in-service training courses and agreed to take part in the research.

The sample size calculation was based on consumption data found in the literature [42,43], ranging from 30 to 60% for the following variables: the percentage of children who ate recommended daily fruits and vegetables; the percentage of children who drank water, based on the recommended daily consumption [44]; and the percentage of children who consumed sweet drinks and sweets during the day. In the absence of information from the literature, the intra-class correlation (ICC) was regarded as 0.3. The sample size for each group (intervention, control), thus, was 20 schools, with 30–35 children per school. This allowed us to identify a change of at least 20% in the consumption of the above-mentioned variables, when the percent of consumption was between 30–60%, α level of 0.05 and with a power of 80% [45].

### 2.3. Measurements

The research methodology had two components: the first dealt with nutrition, employed quasi-experimental comparative methods, and examined the pre-test and post-test changes in the composition of the mid-morning snack and water consumption, brought from home, of the intervention group, compared to the control group. Data were collected by observation of the snack contents before and after the program in both the intervention and the control kindergartens. All observations were made by the teachers.

The second component dealt with physical exercise and its consequences using the students’ feelings following the exercises pictured in the “Paalton” chart (Figure 2), an instrument developed especially for this research. This was a workout chart named from the Hebrew word for “Activity”, designed as an auxiliary tool for the teachers to encourage basic movements suitable for preschoolers [39]. The multi-functional chart with nine easily understood pictures could be used as a game, performed as individual or combined activities, or used by one student to describe correctly an activity for another to perform.

Qualitative interviews documented how the children were able to explain their subjective (body) feelings following physical exercise.

The children in the intervention group were interviewed individually concerning their feelings following the exercises pictured in the “Paalton” chart. Eighty percent of the children responded to the question: How do you feel after the exercise/activity? The children’s responses were recorded by the teachers.

The answers for all of the measures were analyzed using judges from a panel of 12 teachers who did not take part in the intervention, an expert in health promotion, a specialist science teacher for preschool and physical education specialists in early childhood. Scores (in the range of 1–3) comprised the average of two judges, and agreement among them was tested for 364 children (53.6% of respondents); Kappa = 0.812 (*p* < 0.001).

Background data of the research population were collected through questionnaires completed by the teachers, including data related to teacher’s seniority, the number of children in kindergarten, gender, sector and socioeconomic status.

## 3. Results

### 3.1. Data Analysis

Data were collected in both the intervention and comparison groups at the beginning and end of the academic year—September 2011 through June 2012.

Data were analyzed using SPSS version 22, and statistical significance was set at *p* < 0.05. Differences between the intervention and control groups, in the kindergarten and teacher variables, were examined with independent samples *t*-tests and repeated measures analysis of variance. The type of kindergarten and its socio-economic background were analyzed by group using *t*-tests and analyses of variance as appropriate. Repeated measures analysis of variance was used to test the interactions between time and the background variables to identify those that needed to be controlled for.

Differences in the children’s variables were assessed with mixed linear hierarchical models. Children were nested within kindergartens, which violates the assumption of independence of observations. Mixed hierarchical models consider the dependence of observations by giving each sub-group a random effect. Differences between the intervention and comparison groups, in the children’s variables, were thus assessed with mixed models of the variables by time and group, controlling for background variables.

The snacks and sandwich contents were analyzed by group and time using the Z test for differences in percentages. Differences in the quality were analyzed by time (pre-, post-) and group (intervention, control) using mixed linear hierarchical models with repeated measures controlling for the years of experience of the teachers.

### 3.2. Results

Table 2 shows that the experimental and control groups were similar in composition, including data related to teachers’ seniority, the number of children in the kindergarten, gender, sector and socioeconomic status.

Data on the gender of the students were available for 1589 students (84.1%). In 38 of the schools (51.4%), the children were from lower to middle class families; in the remaining 36 schools, the families were of the middle to upper social class. The average experience of the teachers was 14.3 years (SD 9.3).

#### 3.2.1. Pupils’ Nutrition Habits (in Accordance with Previous Publication) [46]

A statistically significant improvement was observed in the composition of the mid-morning snack and in the water drinking habits of the experimental group (Table 3 and Table 4 and Figure 3).

(a) There was a significant increase (44.0% to 58.5%) in the children who brought fruit to school (*p* < 0.001). With regard to vegetables, 25% of the children in the intervention group brought vegetables prior to the intervention and 41.3% did so following it (*p* < 0.001). There were no changes in the comparison group.

(b) There was a significant decline (19.1% to 7.0%) in the percentage of children who brought sweets and snacks to school over the year (*p* < 0.001), with no change in the comparison kindergartens.

(c) There was a significant increase (39.0% to 74.4%) in the percentage of children who brought water to school (*p* < 0.001), compared with 46.9% of the children in the comparison group, which had no change over the school year.

(d) Most children brought sandwiches to kindergarten and there was a significant improvement in their content. A total of 65.7% of the intervention group brought sandwiches with the recommended fillings prior to the intervention (such as cheese, hummus, egg), and 79.8% did so following it (*p* < 0.001). Similarly, there was a significant decline in the percentage of children who brought sandwiches containing sweet spreads such as chocolate or jam, compared to the beginning of the year (from 30.2% to 15.4%, respectively). No changes were observed in the control schools.

#### 3.2.2. Pupils’ Attitudes to Physical Activity

It was found that children were able to express and describe feelings relating to four different aspects of physical exercise after its performance:(1)*Physical aspects*: e.g., heart function, breathing, sweating, thirst, fever, fatigue, difficulty. e.g., easy/not difficult, not tired.(2)A total of 80% of the children referred to the physical aspect (e.g., cardiac activity, perspiration, breathing, heat, fatigue) such as “My heart is beating a lot’’, “My body got hot”, “I felt tiredness in my heart”.*Emotional aspects*: e.g., positive—fun, competence, motivation; negative—not like (do not want/not good/annoying), difficult to manage.A total of 47% of the children expressed themselves in emotional terms. Most of them voiced enjoyment, fun and capability such as: “I can do more”.(3)*Cognitive aspects*: the use of language and concepts of mathematical logic (argument, reason, result), the use of images. A total of 47% of the children made cognitive references that included the use of mathematical-logical concepts and images such as “*many times* it was hard to go up and go down” or “my heart beats *twice as fast*”, sound arguments such as “I had fun *because* I jumped a lot” and cause–effect relationships such as “*if* I jump, *then* my feet hurt”.(4)*Energy expenditure’ aspects*: The responses were classified as follows: 1. Description—a non-scientific explanation, such as “I had a lot of fun”; 2. Scientific description—such as: “my heart is beating strongly:, “I felt heavy breathing”, “It was hard and I was sweating ‘‘, “My blood was flowing fast”, “My body becomes warm”; 3. Scientific description with additional information on enjoyment from the activity—such as: “my heart beats quickly and I want more”, “my body heats up and it was fun”, “I also enjoyed it and I was thirsty”, “I ran and jumped easily, but I’m really sweating”.

A total of 71% of the children offered a scientific explanation of energy expenditure processes resulting from intense physical exercise. Twenty percent of the children also noted the sense of enjoyment that accompanied it.

## 4. Discussion

This study describes how kindergarten children are able to modify significantly their eating and drinking patterns. The changes were reflected in the quality of their mid-morning snack: reduced consumption of sweets and munchies, greater consumption of fruits and vegetables, preference for sandwiches with heathy fillings, and a preference for drinking plain water.

As previously reported by others, kindergarten is a promising setting to change eating habits [47,48]. This study suggests that age-appropriate activities encouraging the use of logical-mathematical thinking enabled the children to understand the desired changes in behavior and to *implement them*. We believe that these changes relate to the self-regulatory abilities that develop during early childhood, including the ability to persevere, delay gratification, to take responsibility and to display self-control. All these qualities are considered to be positively correlated with obesity prevention [49].

Moreover, we have shown that kindergarten children were able to *influence their parents* when they brought from home their mid-morning snacks.

The intervention program did not include the parents directly. The improvement in the composition of snacks which parents prepared at home was probably due to the children’s direct communication with them regarding the teaching of the topic in kindergarten. The study was interventional as we wanted to examine the outcome of an intervention when the children themselves were the main target group. The gap between the parents’ awareness of the need for proper nutrition and the application of this knowledge to the quality of children’s mid-morning snack and water consumption relates to the children having significant skills that can create change in their environment, which, in this case, means their parents’ habitual behavior. Following the training of the teachers, the children conveyed to their parents the facts and guidance they had learnt in kindergarten regarding healthy foods. They also turned this information into a request that their parents prepare more of the recommended foods.

That the children were able to influence their parents to provide them with preferred foods showed that they acted as “agents of change”.

The importance is well known of enlisting the support of parents in children’s dietary changes by, for example, flyers, videos, parent meetings or web sites [3,50,51]. Some intervention programs involve parents directly, being central in aiding their children to acquire early healthful lifestyles, as well as in helping them overcome the reluctance to trying new foods. This can be a significant barrier to adopting healthier dietary habits [3,52]. ToyBox, for example, is a kindergarten-based, family-involved intervention to apply environmental changes and implement lifestyle behaviors to counter obesity [33]. Pediatric weight management interventions in the primary care setting for parents of young children 2–6 years with overweight or obesity resulted in small declines in BMI [53]. Parental involvement is important in health promotion programs for kindergarten children. However, it requires more than involving them actively during the period of an intervention; rather, it concerns developing parental self-efficacy (PSE), which is an attitude that needs to be developed and nurtured as early as possible, even before children are born [54].

Since the intervention program did not directly include parents, the findings show that the improved snack quality was achieved by the children’s direct communication with their parents about their school activities. Thus, the children were indirect and informal *“agents of change”* [55,56]. Yet, it may be assumed that, in order to sustain the results, parent participation must be strengthened over time.

Our study builds on the results of previous intervention studies conducted in Israel [57]. However, instead of concentrating on measuring energy intake, we focused on behavioral outcomes as expressed by the children themselves. Moreover, in our study, physical activity was conducted by the kindergarten teacher and not by an external professional, the importance of which is significant when the intention is to scale up should the program be successful.

It was important to ensure that the participants in the intervention were those who were able to complete the one-year academic training program.

In addition, every kindergarten community is unique, so the training and the intervention were tailored regarding some of the content to the needs of that year. The kindergarten teachers were trained to be sensitive to the way in which the program was applied according to the culture-specific needs of their pupils.

Principles of the Intervention Program—Nutrition

The intervention program was effective in combining continuing in-service education and training for the preschool teachers, incorporating follow-up, and meaningful educational activities through precise and rational language. In contrast to a kindergarten intervention based on public health nurses and dietitians [47,58], our intervention program was implemented by the teachers themselves, independent of any outside authorities.

The intervention program incorporated both *theoretical* and *applied* knowledge. Together, they encouraged the teachers to increase the amount of time they devoted to health and healthy lifestyle promotion, which we consider has led to the improved quality of the food that the children brought from home. They were able to plan illustrative and experiential activities that make use of logical-mathematical thinking and were integrated in the regular daily preschool activities. They created an educational environment that encouraged *self-regulation*—expressed in the ability to *initiate healthy behavior* related to eating fruits and vegetables, to *delay gratification* associated with excessive consumption of sweets, to *choose* to drink water as opposed to sweetened beverages, and to *exercise self-control* with regard to “what and how much I eat and drink”. These traits are all of life-long value. These changes in nutrition were most encouraging, especially in the direction of the Mediterranean diet pattern with increased fruit and vegetable consumption [13,51,59]. Such an effective intervention framework, relying on the teachers, does not require substantial resources on the part of the school and health systems.

Mathematical thinking and related concepts were most relevant when it came to gathering, processing and presenting information within the program. Logical-mathematical thinking enhanced the comprehension of making prudent choices, drawing conclusions and making decisions concerning recommended food, and then implementing them in the mid-morning snack, and in the descriptions of how the children felt after physical exercise—descriptions that were backed by sound arguments and causal explanations

Physical Activity

Regarding physical activity: Low levels of motor skills have been found in overweight compared to normal weight children [60]. This becomes a vicious cycle whereby obese children may refrain from activity and such inactivity contributes to their obesity. Therefore, it is very important to reinforce exercise habits as part of their lifestyle for the rest of their lives.

In this study, we found that preschool children are capable of expressing feelings about their bodies following physical exercise using the “Paalton” chart. In their comments, they referred to physical, emotional and cognitive aspects. Seventy-one percent of the children expressed feelings and physical changes when exerting themselves and made use of scientific concepts that describe “expending energy”. Hence, it is possible to address the subject of energy expenditure by acknowledging what is closest to the child—his/her body. This early appreciation of the physiological phenomena that occur following exercise, and drawing preliminary conclusions regarding the pace of the activity and the amount of energy needed to perform it, are likely to enhance the future understanding of the balance between energy intake and expenditure [32]. However, we found no correlation between energy intake as evidenced by a better-quality mid-morning snack and energy expenditure as recorded by the children’s reactions to physical exercise. The teachers taught concepts of energy input and output, but perhaps each was treated separately with not enough stress on the connection between them. In future intervention programs, more emphasis should be given to illustrate energy balance by using examples such as weighing scales or swings. It is important to treat the human body as a thinking, feeling “machine” that helps children grasp scientific concepts and processes, such as energy balance and metabolic efficiency (at an older age), healthy food choices, as well as how activity affects our emotions (food and mood). Food should be regarded as the “gasoline” for their bodies and should be of the “highest octane”, with regard to portion sizes, variety and balance between major food groups. The aim is for children to self-manage and regulate their behaviors towards healthy growth and development and to prevent overweight and obesity. However, it is probably more important to inculcate long-term healthy lifestyle behaviors, such as active living, rather than focus on weight issues [61], especially when health-related physical fitness (cardiovascular fitness and body mass index) is associated with improved academic performance [62].

A similar program has been carried out in an older girls’ school involving pupils, teachers and mothers. It was successful in improving diet and exercise and has subsequently been scaled up by the Palestinian authority. However, it failed to decrease screen time. This problem needs to be addressed more seriously because of its influence on the development of obesity and poor lifestyle habits [63].

The current intervention emphasized the importance of health education. Health Literacy (HL) is a key outcome of health education and of health promotion efforts [15]. The HL of children and young people is commonly depicted as a multidimensional capability entailing different combinations of individual abilities and knowledge elements. It enables a person to access, understand, judge and act upon health information in relevant and competent ways in order to arrive at health-promoting decisions and actions in different contexts of everyday life [64]. It influences not only their personal health behaviors but also can affect the health actions of their peers, their families and their communities [65]. Furthermore, high HL has been shown to support and protect the influence of mass media on adolescents’ health behavior [66]. Many scholars argue, in accordance with findings from developmental research, that effective HL development can begin in early childhood and that schools are viewed as major settings for its promotion [14]. This includes the development of materials and information that are suited to younger age groups and provided in ways that engage and empower them. Moreover, to achieve HL, more attention needs to be paid to the delivery methods employed in Health Education. Developing teachers’ understanding of constructivist pedagogy and HL will support student’s competency development [67]. Our study may shed light on the possibility of imparting HL already in kindergarten.

In this study, we dealt with three intervention channels. Similar ways to enhance the cooperation between children, parents and the kindergarten staff should be explored in areas such as planning and cooking family meals, holding conversations about food choices, visits to supermarkets and local farmers’ markets, encouraging family activity outings (2 × 2, rather than 4 × 4—i.e., hiking more than riding in jeeps), and paying more attention to how children express themselves, especially after exercise. Further work is required to ensure the sustainability and scale-up of such programs. As this study shows, young children can understand, take responsibility, change behavior, and be agents of change and leaders of healthy behaviors. This is a very important message for parents and decision makers that determine educational policy.

## 5. Limitations

The present research was conducted over the period of a single school year, and it remains to be seen whether the behavioral changes observed are sustained and whether there are effects on body weight. The maintenance of new habits requires a long-term plan. *A priori*, the research refrained from creating situations that were potentially sensitive to the teachers or threatening to the children and their parents, such as body weight and BMI measurements. These require parental consent but do provide the “hard” evidence for successful prevention as measured in many intervention programs, even though there are inconsistent results [28,33].

Referring to the study population, it should be noted that the division into groups was not conducted randomly because the intervention plan was based on in-service training for kindergarten teachers. This may have introduced a selection bias. However, as shown, the two groups were similar in background variables with regard to: number of children in class, gender, socio-economic status of the school population, affiliation of kindergarten, and experience of the teachers.

Regarding the data collection, there could have been an intent (obsequious) bias by potential interpretation, but this was mitigated, in part, by the semi-structured nature of the questionnaire. In addition, data were anonymized during analysis and the kindergarten teachers knew this. Perhaps this also helped to mitigate the potential bias.

## 6. Conclusions

This intervention study on changing (improving) lifestyle habits has been effective at three levels—directly for the teachers’ attitudes and degree of involvement in the subject, for the children, and indirectly for their parents. An outcome of this study is the use, by the Ministry of Education, of the tools we have developed to promote 21st century skills by aligning future educational curricula with the acquisition of healthy behaviors and lifestyles.

The next stage of this intervention can include: similar interventions provided by the task force for reducing obesity [12]; designing future research to allow long-term follow-up; applying the changes in the quality snacks to the meals at home; improving meal composition among other family members; application of the intervention in therapeutic settings; building an intervention program that presents the connection and balance between energy input and energy output through experiences and age-compatible illustrations; and expanding the principles of the intervention to including behaviors that contribute to a healthy lifestyle—for example, controlled exposure to the sun, using sunscreen, reducing sedentary activity, and maintaining oral and dental health.

However, we believe that appropriate programs should be designed to give continuity in educating healthy lifestyles for the prevention of obesity *throughout* the school years. Future research, as the current study indicates, should consider developing leadership abilities for a healthy lifestyle in children at an early age, on a personal level, and within the family and community settings—and the earlier they begin, the better.

## Figures and Tables

**Figure 1 nutrients-15-02615-f001:**
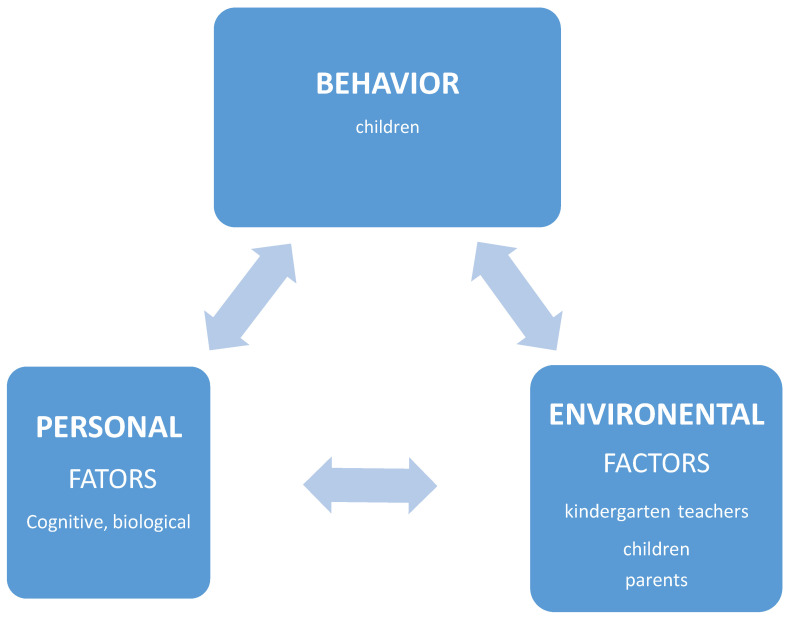
The research model in the context of the Social Cognitive Learning Theory.

**Figure 2 nutrients-15-02615-f002:**
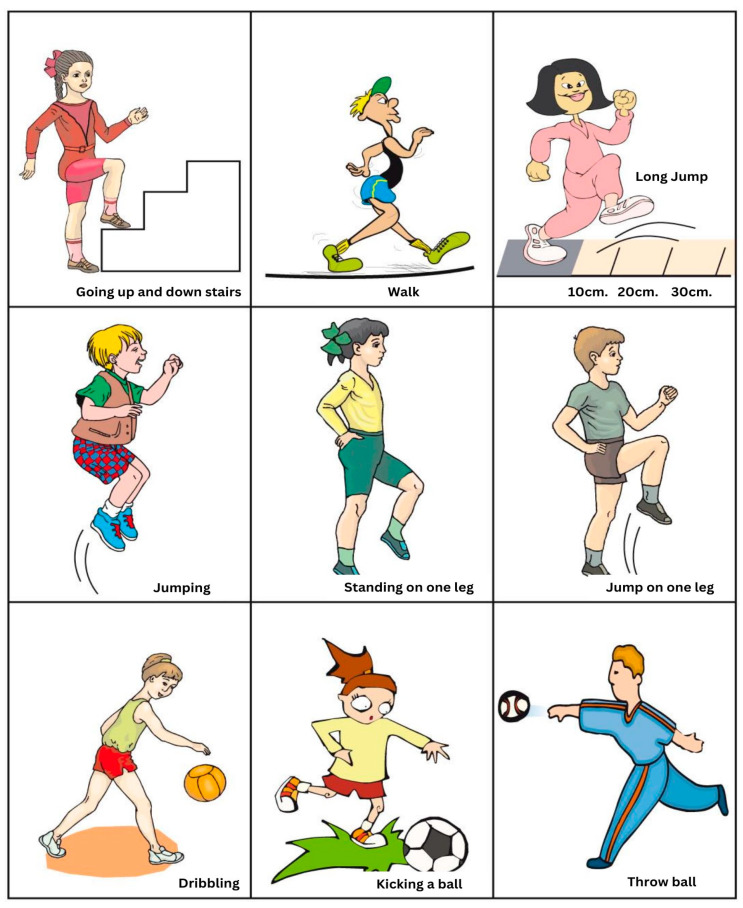
The “Paalton” chart for movement skills among kindergarten children.

**Figure 3 nutrients-15-02615-f003:**
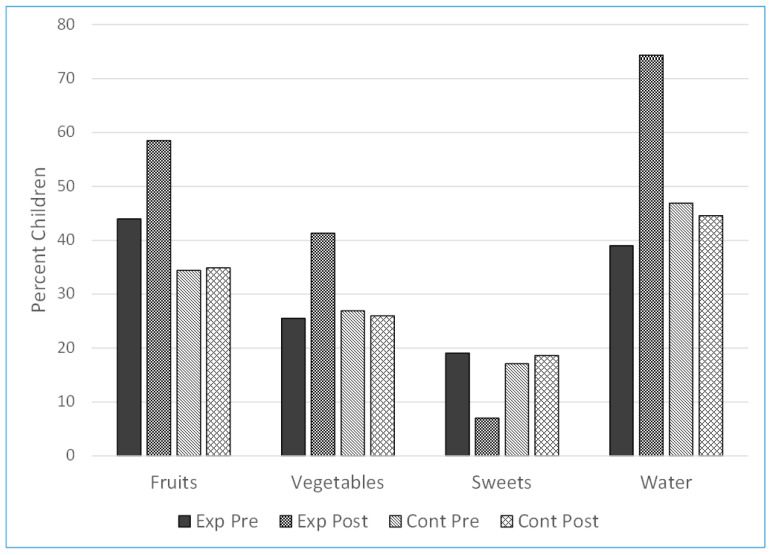
The effect of the intervention program on the consumption of Fruits, Vegetable, Sweets and Water.

**Table 1 nutrients-15-02615-t001:** Lifestyle intervention program for kindergarten teachers.

Theme	Topics
Health Education and Promotion	1 Obesity epidemic, causes 2 Infant Growth and development: physical, cognitive and social 3 Healthy diets and their effects on learning and development 4 Health education and the role of the kindergarten for health promotion 5 Energy balance
Nutrition	1 The critical position of nutrition in growth 2 Macro and micronutrients: The role of Proteins, Carbohydrates, Fats, Fiber and Vitamins 3 The food pyramid (Mediterranean diet); design characteristics—food groups, portions, variety, balance 4 Food choices and self-control 5 The importance of water and fluid balance
Physical Activity	1 Role of activity (exercise) in healthy growth and in the prevention of chronic non-communicable diseases 2 Activity and energy balance 3 New methods for encouraging activity in the kindergarten 4 The “Paalton” exercise diagram and its use with mathematical language 5 Expressing feelings after physical activity
Imparting skills for logical and mathematical thinking	1 Using numbers: counting, comparisons, recognizing and documenting quantities 2 Quantifying in daily life: concepts of size and proportions, estimating amounts—length and volume, concepts of time, calendar days, weeks and months 3 Using charts for records in the school 4 Thinking skills: (a) recognition, understanding, summarizing, comparisons, organization, estimation and measurement, collecting and representing data using charts, graphs; (b) choice through justification, self-criticism, reaching conclusions, problem solving, logical connections between cause and effects, “If…. then…” 5 Combining health issues with mathematical language and thinking skills concerning physical exertion during the year in relation to time (holidays, festivals) and place (home, kindergarten, playgrounds)

**Table 2 nutrients-15-02615-t002:** Characteristics of the study populations.

Kindergartens	Intervention Group (42)		Comparison Group (32)		
	Percent (SD)		Percent (SD)		
Number of Children				842	1048
Boys	55.7	NS *	56.3	454	436
Girls	44.3	NS	43.7	352	347
Number of children in class	(6.2)	NS	(7.5)	26.3	25.0
Experience of the teachers (year)	(9.2)	NS	(9.7)	14.0	14.5
Sector of Kindergarten					
National—Secular	47.6		75.0	24	20
National—Religious	52.4		25.0	8	22
Socio-Economic Status of school population					
Low—Low-Middle	47.6		56.3	18	20
High-middle—High	52.4		43.8	14	22

* NS = Not significant.

**Table 3 nutrients-15-02615-t003:** Description of the contents of the mid-morning sandwich.

	Intervention Group			Comparison Group		
Sandwich Content	Difference in %	post *N* = 653	pre *N* = 740	Difference in %	post *N* = 730	pre *N* = 734
	Change%	*N* (%)	*N* (%)	Change %	*N* (%)	*N* (%)
Cheese	+6.1 **	273 * (41.8)	264 (35.7)	+0.5	221 (30.3)	219 (29.8)
Chocolate Spread	−12.0 ***	64 ** (9.8)	161 (21.8)	+0.2	(5.1)	(4.9)
Humus	+3.6 **	102 * (15.6)	89 (12.0)	+2.0	87 (11.9)	73 (9.9)
Egg	+0.5	53 (8.1)	56 (7.6)	+0.8	58 (7.9)	52 (7.1)
Salami	+1.4	37 (5.7)	32 (4.3)	−1.9	51 (7.0)	65 (8.9)
Tuna	+2.7 **	47 * (7.2)	33 (4.5)	−1.2	36 (4.9)	45 (6.1)

* *p* < 0.05, ** *p* < 0.01, *** *p* < 0.001.

**Table 4 nutrients-15-02615-t004:** Breakdown of the contents of the mid-morning sandwich according to its quality.

	Intervention Group			Comparison Group		
	Difference %	post *N* = 648	pre *N* = 728	Difference%	post *N* = 719	pre *N* = 722
		*N* (%)	*N* (%)		*N* (%)	*N* (%)
Bread only—no filling	+0.7	31 (4.8)	30 (4.1)	+0.7	35 (4.9)	30 (4.2)
Preferred (healthy) content	+14.1 ***	517 (79.8)	478 (65.7)	+0.4	460 (63.9)	459 (63.5)
Sweet spreads	−14.8 ***	100 (15.4)	220 (30.2)	−1.1	224 (31.2)	233 (32.3)

*** *p* < 0.001.

## Data Availability

The data are available for R.J.

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
