# Peer review of "Developing Healthy Lifestyle Behaviors in Early Age—An Intervention Study in Kindergartens"

_nutrients, 2023, doi:10.3390/nu15112615_

Round 1

Reviewer 1 Report

Thank you very much for giving me the opportunity to revise this interesting manuscript by Jakobovich et al.  on the intervention study at kindergarten to develop healthy lifestyles with children. The manuscript is well written, and the topic is timely, as underlined by the Authors. There is only one thing I recommend to the authors (and other minor issues), and it's to better specify contribution of lifestyle changes by the indirect involvement of parents. What about their degree of involvement, parental health literacy and home environment? While much focus has been given to first two aims, less attention has been given to the third one. 

The final sample for analysis should be placed on the results and not at the methods.

Did the authors apply any exclusion criteria for the participants (kindergarten children)?

Data were collected by observation and all observations were made by the teachers. They also conducted cognitive interviews Are they trained for collecting data in any way? Authors should provide comprehensive information on how they could guarantee for the validity of the process of observation.

The topic under discussion is very important and will have relevance for the health policy makers. However, the proposed discussion lacks a comparison of the results obtained in this and other studies.

Also, I would recommend to the authors to add an explanation of the samples representative character.

In table 2 acronym NS should be described below the table.

Author Response

1.Better specify contribution of lifestyle changes by the indirect involvement of parents. What about their degree of involvement, parental health literacy and home environment? While much focus has been given to first two aims, less attention has been given to the third one. 

Thank you for this important comment. We added an explanation in the paragraph  "Discussion" lines 330-343

The intervention program did not include the parents directly. The improvement in the composition of the snacks which the parents prepared at home was probably due to the children's direct communication with their parents regarding the teaching of the topic in kindergarten. The study was interventional as we wanted to examine the outcome of an intervention where the children themselves were the main target group. The gap between the parents’ awareness of the need for proper nutrition and the application of this knowledge to the quality of children's mid-morning snack and water consumption relates to the children having significant skills that can create change in their environment, which, in this case, means their parents’ habitual behaviour. Following the training of the teachers, the childrens conveyed to their parents the information and guidance they had acquired in kindergarten regarding healthy foods. They also turned this information to request that their parents prepared more of the recommended foods.

 That the children were able to influence their parents to provide them with preferred foods showed that they acted as "agents of change".

  1. Regarding aim three of the study please see lines 120-122                        The indirect parent involvement through the children was shown by changing the quality of the children's mid-morning snack and water consumption – thus children were acting   as “agents of change”.

3.The final sample for analysis should be placed on the results and not at the methods.

Thank you. We made the change, please see line 237.

4.Did the authors apply any exclusion criteria for the participants (kindergarten children)?

All children were included in the kindergarten participation study as this was a real life intervention.

5.Data were collected by observation and all observations were made by the teachers. They also conducted cognitive interviews Are they trained for collecting data in any way? Authors should provide comprehensive information on how they could guarantee for the validity of the process of observation.

The teachers indeed received 8 hours of  special training for conducting observations and collection data dedicated specifically to the skills development.  Importantly,  additional training, included a simulation for this throughout the training supervised by the researchers - two of the authors on this manuscript.

In addition ,we created a tool as specifically as possible,  that would be easy to use with no need for interpretation  (For example, yes/no question

about chocolate spread)

*Please see also the comment added in" Limitations" paragraph  lines 493-496

Regarding Data collection, There could have been an intent (obsequious) bias by potential interpretation but this was mitigated, in part, by the semi-structured nature of the questionnaire. In addition, data were  anonymized during analysis and the kindergarten teachers knew this. Perhaps this also helped to mitigate the potential bias.

  1. The topic under discussion is very important and will have relevance for the health policy makers. However, the proposed discussion lacks a comparison of the results obtained in this and other studies.

Thank you for the comment. We would like to note that this action intervention was unique in its nature as it targeted the children in specifical intervention which included active road of kindergarten teachers. The intervention evaluated in this study was different because it was conducted by teachers nad not health professionals health professionals who have inherent training in this topic in their basic studies so it is not very comparable. We did our best to compare but there aren’t other studies  to compare it to .

*Please see also the added explanation  in" Introduction" paragraph  lines 87-99 that refers to studies in the literature and what is lacking, providing the basis for our study.

Childhood obesity has catalyzed many important studies, many of which focus on measurement tools for body weight, their validation and fidelity (23) as outcome measures , Yet the importance of observing health behaviors and, moreover, analyzing children’s verbal responses are critical for capturing their experiences and the significance that they expressed regarding participating in the intervention.

Programs have been tested that are based on the role of the kindergarten teacher in conducting the intervention focusing on physical activity and motor ability outcomes  while also identifying gender and socio-economic associations with positive responses to the activity(24)(25).

Learning of children’s attitudes towards healthy eating is of utmost importance, even when the intervention includes providing directly nutritious foods (26).

And the comment added in" Discussion" paragraph  lines 356- 360; 368-374

Parental involvement is important in health promotion programs for kindergarten children. However, it requires more than involving them actively during the period of an intervention, rather it concerns developing parental self-efficacy (PSE) -  that is an attitude that needs to be developed and nurtured as early as possible, even before children are born (54).

Our study builds on the results of previous intervention studies conducted in Israel (57); however, instead of concentrating on measuring energy intake, we focused on behavioral outcomes as expressed by the children themselves. Also, in our study, physical activity was conducted by the kindergarten teacher and not by an external professional, the importance of which is significant when the intention is to scale up should the program be successful.

6.To add an explanation of the samples representative character

This research  included all children that  attended kindergartens from all areas of the the wider Jerusalem District of the Ministry of Education (Jerusalem municipal plus towns and settlements in the surrounding  from all of these areas)

  1. 7. In table 2 acronym NS should be described below the table

Thank you. We added the acronym that was missing.

Please see line  539

Reviewer 2 Report

This is an intervention study to introduce healthier behaviour using the setting of the kindergartens. Data is very old especially taking into account the large body of research in the same area and with similar methods. It is not clear why the authors decided to publish the results now, 10 years after the evaluation of the intervention. Because of that, references are rather old and thus not thoroughly presenting nowadays knowledge. Follow-up data about the rate of children with sustainable knowledge and behaviour.

There are also concerns about the randomization of the kindergartens based on the choice of the kindergarten teachers.

Author Response

1.This is an intervention study to introduce healthier behaviour using the setting of the kindergartens. Data is very old especially taking into account the large body of research in the same area and with similar methods. It is not clear why the authors decided to publish the results now, 10 years after the evaluation of the intervention. Because of that, references are rather old and thus not thoroughly presenting nowadays knowledge. Follow-up data about the rate of children with sustainable knowledge and behaviour.

  The first paper on this study was published in 2019 (Jakobovich et al ,2019 ). Over the past decade, the topic has become increasingly relevant and a growing need for studies such as this to find the best interventions that can be used in real life situations. Therefore, we have added more detailed information on the effects of physical activity to our initial publication of the intervention study

Although the intervention in the current study took place in the kindergarten setting and continued at home, the emphasis was on the  children, themselves,  as they switched over to ‘doing’ and with the cooperation of their parents. . This changing behavior occurred on the basis of the children's understanding the expected behavioral change , while connecting relevant knowledge and skills of cultivating mathematical logical thinking and language- facilitating their roles as  "agents of change".

These educational skills we used in our intervention methods have been discussed recently (Hochschul et al,  2021) to  include: taking responsibility, problem solving, , getting others to cooperate in the process of change, building a solid literacy and numeracy foundation- and becoming change agents – all these skills as the best preparations for the future.

2.There are also concerns about the randomization of the kindergartens based on the choice of the kindergarten teachers

Thank you  for the comment. We added an explanation in the "Discussion" paragraph in lines 375-381

It was important to ensure that the participants in the intervention were those who were able to complete the 1-academic year training program.

In addition, every kindergarten community is unique so that the training and the intervention was tailored regarding some of the content to the needs of that year. The kindergarten teachers were trained to be sensitive to the way in which the program was applied according to culture- specific needs of their pupils.

Reviewer 3 Report

Overall, a good read. Few issues to consider listed below.

1. Please provide evidence for intervention components that are directed to Kindergarten children (lines 146-156)

2. Presentation of qualitative study findings can be improved - % of children responding to a certain feeling is informative but shouldn't be the only focus. Further discussion regarding those feelings or variety of feelings should be discussed. Listing all the quotes is also not necessary.

3. Lines 300-314; dismissing parental or family involvement in lifestyle interventions is rather a strong argument. There are plenty of family based interventions, especially among young children that have shown positive changes.

4. As noted in 'limitations', the effect of this intervention need to be investigated post intervention phase. Any plans to progress to the next stage could be included in 'conclusion'.

Author Response

1.Please provide evidence for intervention components that are directed to Kindergarten children (lines 146-156)

Thank you for this important and helpful  comment.

Please see the comment added in" Introduction" paragraph  lines  54-57, 87-99

The recommendations for childhood nutrition, physical activity and reduction of screen-time are as relevant as ever as expressed by the expert teams in the Position statement on Childhood Obesity from the Global Federation of International Societies of Paediatric Gastroenterology, Hepatology and Nutrition (FISPGHAN) (12)

Childhood obesity has catalyzed many important studies, many of which focus on measurement tools for body weight, their validation and fidelity (23) as outcome measures, Yet the importance of observing health behaviors and, moreover, analyzing children’s verbal responses are critical for capturing their experiences and the significance that they expressed regarding participating in the intervention.

Programs have been tested that are based on the role of the kindergarten teacher in conducting the intervention focusing on physical activity and motor ability outcomes  while also identifying gender and socio-economic associations with positive responses to the activity (24)(25).

Learning of  children’s attitudes towards healthy eating is of utmost importance, even when the intervention includes providing directly nutritious foods (26).

And the comment added in" Discussion" paragraph  lines 368 -374

Our study builds on the results of previous intervention studies conducted in Israel (57); however, instead of concentrating on measuring energy intake, we focused on behavioral outcomes as expressed by the children themselves. Also, in our study, physical activity was conducted by the kindergarten teacher and not by an external professional, the importance of which is significant when the intention is to scale up should the program be successful.

  1. Presentation of qualitative study findings can be improved - % of children responding to a certain feeling is informative but shouldn't be the only focus. Further discussion regarding those feelings or variety of feelings should be discussed. Listing all the quotes is also not necessary.

A qualitative tool was included  for this part of the research, and therefore quotes are given to help the readers understand the types of feelings that were expressed. However we reduced the number  of the selected quotes.

Please see paragraph  3.2.2 - "Pupils’ attitudes to Physical Activity"    lines 286 -314

3.2.2. Pupils’ attitudes to Physical Activity

It was found that children are able to express and describe feelings relating to four different aspects of physical exercise after its performance:

1) Physical aspects: e.g., heart function, breathing, sweating, thirst, fever, fatigue, difficulty. e.g. - easy / not difficult, not tired.

 80% of the children referred to the physical aspect (e.g. cardiac activity, perspiration, breathing, heat, fatigue) such as “My heart is beating a lot'', "My body got hot", “I felt tiredness in my heart”;

2) Emotional aspects: e.g. positive - fun, competence, motivation; negative - not like (do not want / not good / annoying), difficult to manage.

47% of the children expressed themselves in emotional terms. Most of them expressed enjoyment, fun and capability such as: “I can do more".

3) Cognitive aspects: the use of language and concepts of mathematical logic (argument, reason, result), the use of images. 47% of the children made cognitive references that included use of mathematical-logical concepts and images such as "many times it was hard to go up and go down" or "my heart beats twice as fast", sound arguments such as "I had fun because I jumped a lot" and cause-effect relationships such as "if I jump, then my feet hurt".

4) 'Energy expenditure' aspects: The responses were classified as follows:                     1. Description a non-scientific- explanation such as, “I had a lot of fun”; 2. Description scientific - such as: "my heart is beating strongly:, “I felt heavy breathing”, "It was hard and I was sweating '', “My blood was flowing fast”, “My body becomes warm”; 3. Description scientific with additional information on enjoyment from the activity -such as:   'my heart beats quickly and I want more”, “my body heats up and it was fun”, “I also enjoyed it and I was thirsty” , “I ran and jumped easily, but I'm really sweating”.

71% of the children offered a scientific explanation of energy expenditure processes resulting from intense physical exercise. Twenty percent of the children also noted the sense of enjoyment that accompanied it.

  1. Lines 300-314; dismissing parental or family involvement in lifestyle interventions is rather a strong argument. There are plenty of family based interventions, especially among young children that have shown positive changes.

Thank you for the important comment that raises the need  to refine and clarify  what is written.

There was no decision to dismiss or include parents. Thus, we did not instruct the kindergarten teachers not to mention or talk about the program to the parents. The focus was principally on the children - to learn about lifestyle and what messages would be brought home regarding mid-morning  snack and water consumption.

Moreover, children were given exercises to conduct at home together with their parents Thus, children were not only involved directly, they were also encouraged to include their parents and even given tools to monitor their changes at home with the support of the parents. The improved quality of the mid-morning snack derived from the children's direct communication with their parents about their activities at kindergarten - children acting as  informal "agents of change."

4. As noted in 'limitations', the effect of this intervention need to be investigated post intervention phase. Any plans to progress to the next stage could be included in 'conclusion'.

Thank you very much for emphasizing the need to address the next stages.  We have addressed this in the paragraph of "conclusions" in the lines 503-519

The tools that we have used in this program are now being used by the Ministry of Education to acquire 21st century skills.

The next stages of this intervention can include: Similar interventions provided by the task force for reducing obesity; Designing future research to allow long term follow-up; Applying the changes in the quality snacks to additional meals at home; Improving meal composition among other family members; Application of the intervention in therapeutic settings; Building an intervention program that presents the connection and balance between energy input and energy output through experiences and age-compatible illustrations; Expanding the principles of the intervention to additional behaviors that contribute to a healthy lifestyle - for example, controlled exposure to the sun, using sun screens, reducing sedentary activity; and maintaining oral and dental health. However, we believe that appropriate programs should be designed to give continuity in educating healthy lifestyles for the prevention of obesity throughout the school years. Future research, as the current study indicates, should be taken into account developing leadership abilities for a healthy lifestyle in children at an early age on a personal level and within the family and community settings-and the earlier they begin the better.

Round 2

Reviewer 2 Report

Dear Authors,

Thank you for the substantial improvement of the manuscript especially in the methodological part. Although your findings are not original, they could of merit to the scientific and general public.

Author Response

Thank you for the  comments.

We addressed  the minor linguistic issues.